# Subcranial Encephalic Temnograph-Shaped Helmet for Brain Stroke Monitoring

**DOI:** 10.3390/s24092887

**Published:** 2024-04-30

**Authors:** Antonio Cuccaro, Angela Dell’Aversano, Bruno Basile, Maria Antonia Maisto, Raffaele Solimene

**Affiliations:** 1Department of Informatics, Modeling, Electronics and Systems Engineering (DIMES), University of Calabria, 87036 Rende, Italy; antonio.cuccaro@unical.it; 2Department of Engineering, Università degli Studi della Campania Luigi Vanvitelli, 81031 Aversa, Italy; angela.dellaversano@unicampania.it (A.D.); raffaele.solimene@unicampania.it (R.S.); 3TTC Medical Srl, 80013 Casalnuovo di Napoli, Italy; bruno.basile@temnografia.com

**Keywords:** stroke detection, MUSIC algorithm, incoherent approach

## Abstract

In this contribution, a wearable microwave imaging system for real-time monitoring of brain stroke in the post-acute stage is described and validated. The system exploits multistatic/multifrequency (only 50 frequency samples) data collected via a low-cost and low-complexity architecture. Data are collected by an array of only 16 antennas moved by pneumatic system. Phantoms, built from ABS material and filled with appropriate Triton X-100-based mixtures to mimic the different head human tissues, are employed for the experiments. The microwave system exploits the differential scattering measures and the Incoherent MUSIC algorithm to provide a 3D image of the region under investigation. The shown results, although preliminary, confirm the potential of the proposed microwave system in providing reliable results, including for targets whose evolution is as small as 16 mL in volume.

## 1. Introduction

In brain stroke, “timeliness” is the operative word and, indeed, it is necessary to act quickly. Specifically, one needs to act within 60 min (the so-called door-to-needle Golden Hour) [1] to avoid death or permanent disability in patients. In [2], it is shown that every minute in which a large vessel ischemic stroke is untreated, the average patient loses 1.9 million neurons, 13.8 billion synapses, and 12 km (7 miles) of axonal fibers. Each hour in which treatment fails to occur, the brain loses as many neurons as it does in almost 3.6 years of normal aging. Therefore, on-the-spot and rapid diagnosis is crucial to reduce the death of many brain cells, with a resulting improvement in faster post-treatment convalescence.

Conventional technologies, such as magnetic resonance imaging (MRI) and X-ray computed tomography (CT), can only be used in hospitals. CT uses ionizing radiation and MRI is very expensive. Moreover, continuous monitoring is not imaginable. Recently, microwave imaging has attracted attention from researchers as a valid diagnostic solution. It is an alternative to the above-mentioned ones thanks to its non-ionizing, low-power, non-invasive, potentially low-cost, and portable features.

The microwave techniques are based on the existence of a dielectric contrast between different tissues. By inspecting several studies on the dielectric properties of different human tissues, a clear indication of a significant contrast between blood and white and gray brain matter can be deduced [3]. This provides the basis for detecting the pool of blood due to a hemorrhagic stroke by microwave approaches.

In the literature, different devices are presented. The first prototype was proposed by Chalmers University of Technology in Göteborg (Sweden) and Medfield Diagnostics AB [4]. This system consists of 12 Tx/Rx triangular patch antennas with small plastic balloons filled with water as a matching medium. The stroke classification (between hemorrhagic and ischemic stroke) is performed by measuring the whole scattering matrix, over the frequency range [0.8–1.3] GHz, and processing the data through a machine-learning algorithm. Another system is shown in [5], where 177 antennas are used as transmitters and receivers. This system operates at a single frequency of 1 GHz, and the imaging chamber is filled with a matching liquid. The reconstruction time is on the order of 2 min using a parallel computer with 4096 cores for a 2D image [6]. The results are obtained from noisy synthetic data from a very accurate model of the brain. However, this system shows disadvantages in terms of cost and portability. Another device developed in [7], where a radar-based system with 16 antennas in monostatic configuration is used and works in the frequency band [1–2.4] GHz. Here, the matching medium is not employed but the antennas are kept at a fixed distance from the head. In this case, the antenna arrangement does not allow for a compact structure in terms of saving space. These are only three of several examples shown in the literature in recent years [8,9,10,11,12]. The main difference between them concerns the developed reconstruction algorithm and the antenna design. The role played by them is very crucial and can affect the successful detection of a stroke. In this paper, a compact helmet-shaped microwave system for stroke detection, named TES (Subcranial Encephalic Temnograph), is presented. Compared to the literature, the proposed system exploits a differential Incoherent MUSIC algorithm [13]. The approach consists of collecting data at two different instants, $t_{1}$ and $t_{2}$, building the multiview multistatic matrix of difference data, and evaluating the pseudospectrum at each frequency. The multiple frequencies are incoherently combined by factorizing all pseudospectrums. Finally, 3D stroke detection is achieved by a slice approach as a collection of 2D pseudospectrums at different heights. The considered strategy has two main advantages. First, since it is a differential strategy, clutter rejection is no longer required and the system can work without a matching medium, positively affecting its portability. Second, an incoherent combination allows one to achieve stable detection results against a lack of tissues’ frequency response information. The preliminary release of such a system was presented in [14]. However, herein, the TES system is further improved. In particular, it is optimized in terms of its simplicity, rapidity of use, and compactness, which are key factors in emergency situations. This requires reviewing some parts of the system, such as the RF block and the antenna, that are designed to work in direct contact with the head. This avoids the matching medium and positively affects the system’s portability. In addition, a different and enriched performance analysis of the system is addressed. This paper has the following structure. In Section 2 and Section 3, the prototype is described in all its parts. In Section 4, the complex heterogeneous phantom composed of the skull, cerebrospinal fluid (CSF), brain, and skin layer is shown. In Section 5, the algorithm for brain stroke localization is described and is validated by experimental results in Section 6. The Conclusion ends this paper.

## 2. TES Detection System

In this section, the prototype microwave system shown in Figure 1 is described. As can be seen, the idea is to scan the head using a helmet containing *N* antennas by adopting a multiview, multistatic configuration. In particular, the three main parts of the architecture system, the RF board, the helmet hosting the antennas, and the pneumatic system, are summarized in Figure 2.

### 2.1. RF Circuit

As in any microwave imaging system, establishing the appropriate frequency band is crucial. In brain imaging applications, a good trade-off between penetration depth and resolution is desirable. This can be achieved by selecting the frequency band [1–2] GHz [15]. Accordingly, the entire RF block will be chosen in order to work in this frequency range.

The RF block is a standard transceiver system operating in the bandwidth of interest. In order to lighten the helmet, the transmitter (TX) based on a PLL, the variable-gain amplifier (VGA), and the receiver (RX) block based on a direct conversion (DC) were assembled into a separate box connected to the helmet via 1.5 m long cables. All these RF elements are off-the-shelf components (Linear Technology circuits). Differently, the switch matrix was designed in order to mount it on the helmet. Accordingly, the criteria of compactness and circuit simplicity were followed. This last was achieved by considering only the paths associated with the transmitting mode ($S_{21}$), then renouncing to the reflection coefficient measurements ($S_{11}$). Overall, the switch matrix was composed of 4 eight-channel switches (8SW), 16 two-channel switch antennas, and 2 two-channel switches (2SW). In order to reduce the overall matrix size and satisfy the compactness constraint, all switch matrix circuits were homemade.

### 2.2. Pneumatic System

The system allows one to control the movements (toward and away from the head) of each antenna individually. This ensures their correct placement on the head, avoiding air-gap situations. In detail, there are sixteen flexible pipes connected to micro-pistons. Each piston moves a single antenna gradually and generates a maximum mechanical force of about 0.2 N when the radiator touches the skin tissue. This improves the measurement stability since it avoids unwanted movements of the antenna during the acquisition process. This is crucial if a differential strategy is adopted [16].

### 2.3. Helmet Design

In this section, the material used for helmet realization is discussed. Among the different materials, the one most suitable for making objects with a 3D printer is ABS (Acrylonitrile Butadiene Styrene) [17]. This is due to its low price, high mechanical strength, and excellent electrical insulation properties [18,19,20,21,22,23]. Accordingly, both the external (Figure 1) and internal (Figure 3) structure of the helmet are made of ABS. In particular, the inside area has many square holes all around its surface, into which sixteen pneumatic pistons are wedged. The antennas are anchored to the end of the pistons in order to provide complete and uniform head scanning. Additionally, the outer shell is completed with two control buttons used for device functioning. The orange button turns the device on/off, while the other blue button enables the measurement and the processing of data (see Figure 1). As can be seen from Figure 1, to facilitate the user, the helmet is equipped with two strip-led indicators. The white color defines the scanning stage (panel (a)); the blue color identifies the processing data (panel (b)), which provides binary results: target detected or not detected. In the first case, a red color appears; conversely, the strip-led emits a green color.

## 3. Antennas

Generally, the penetration depth in human tissues is limited [24]. This can be improved by immersing the antenna into coupling medium. Another advantage of using a coupling medium is to dramatically reduce the physical size of the antenna since this works in a medium with lower λ compared to free space. Accordingly, this seems to be a mandatory constraint from the perspective of a portable and compact device. Obviously, this greatly complicates the feasibility of the prototype and its effective portability. Indeed, one should imagine a tank containing liquid (typically, a mixture of water and Triton X-100 [25] or an oil-in-gelatine mixture [26], used for the representation of human tissues, can be adopted) where the antennas are drowned, with all the critical issues on how to feed the radiators passing from free space to the coupling medium. Everything becomes simpler when the antenna is positioned directly on the body. Indeed, the advantages achieved by a coupling medium are preserved without it.

In [27], a compact 4×4 wide band array is proposed when it is directly in contact with the skin. Following the same concept, a planar structure is proposed in [28] or, for instance, a flexible array system is proposed in [29].

According to [30,31,32], where the radiation of an antenna aperture is studied in detail, a wide slot antenna is considered for our purposes. This solution matches the compactness and lightness purpose of the entire diagnostic system. The antenna design takes as a starting point a slot antenna printed on a substrate of standard FR4 (thickness $h_{s}=0.8$ mm, $\epsilon_{r}=4.4$, and σ=0.012 S/m), which was initially designed to work in contact with the simplified semi-infinite head phantom shown in Figure 4a. This choice has a two-fold motivation. The first is that the compactness of the system requires a constraint on small antenna sizes; accordingly, the head interface can be considered locally flat. In addition, the first layers of a human head are skin and the skull. Initially, both skin and skull tissues are considered homogeneous and lossless. The skin thickness, $t_{skin}=2$ mm, with permittivity, $\epsilon_{r_{skin}}=36$, and the skull layer with permittivity, $\epsilon_{r_{skull}}=15$, and very large thickness, 150 mm, are taken into account.

Basically, the proposed antenna can be schematized in two elements: the ground plane hosting the wide slot and the microstrip feed line with a fork-like tuning stub. By properly selecting the parameters of the fork-feed, coupling between the microstrip and wide slot can be controlled, thus providing enhancement of the operative bandwidth. The prototype is shown in Figure 4b,c. The antenna was optimized to combine a compact size and a minimum −10 dB return loss across the operating band [1–2] GHz, which guarantees a good trade-off between the electromagnetic (EM) penetration inside the head tissues and the spatial imaging resolution.

Moreover, to facilitate the antenna positioning on the human head, a micro-SMA connector was placed on the side of the fork-feed. The final geometrical dimensions of the designed slot antenna are listed in Table 1. Note that the area of the front (radiating) part of the proposed slot antenna is very small; it is about 4 cm^2^. This facilitates the positioning of the N=16 antennas inside the TES prototype.

According to the studies carried out in [30,31,32], the slot antenna is strongly affected by the loading effect of the phantom. This can be appreciated by considering Figure 5b. Here, the behavior of $S_{11}$ in terms of frequency is shown within the selected operative band when the antenna is in contact with different phantoms. In particular, the blue line refers to the two-layered phantom used during the design procedure (Figure 4a—lossless case). In the same figure, the red line is still relative to a two-layered phantom but losses in skin and the skull tissues are included. Finally, the green line refers to the phantom depicted in Figure 5a. This more complex phantom, named complete phantom, consists of five layers mimicking skin, skull, cerebrospinal fluid (CSF), white matter, and gray matter tissues. The properties of the tissues were taken from the literature [3] and are reported in Table 2 at the frequency of 1 GHz. Note that the tissue properties (i.e., thickness and dielectric properties) of the complete phantom are different from those of the simplified phantom adopted in the design procedure. As can be seen, the loading effects of the phantom change the resonance distribution of $S_{11}$ in the frequency band of interest. However, what is relevant is that the constraint on the return loss holds. In fact, in all the cases $S_{11}$ is below −10 dB, practically across the whole operative frequency band. Finally, Figure 6 reveals a good EM wave penetration into the complete phantom at a frequency of 1 GHz.

Certainly, a critical aspect of a slot antenna is its placement on the human head. Accordingly, a first analysis takes into account the impact of introducing a small air-gap between the slot antenna and the head surface. The antenna reflection coefficient $S_{11}$ is evaluated by varying the air-gap between the antenna and the skin surface of the phantom in Figure 5a. The results are shown in Figure 7. As can be seen, the antenna performance drops sharply if the perfect contact fails. However, these limits are overcome by using a pneumatic system, as shown in Figure 2, which avoids air-gap circumstances. For the management of the subsystems, a Host PC with the Windows 10 operating system was used.

## 4. Phantom Realization

In order to validate the prototype system, a phantom that mimics the electromagnetic characteristics of the head was built. Four different tissues (bone, CSF, brain, and blood) were realized following the procedure reported in [25], where each head tissue is a liquid mixture based upon Triton X-100 and salted water. Table 3 summarizes their recipes and the electromagnetic characteristics following the Cole–Cole model at 1 GHz. In our phantom, the brain was considered a blend of white and gray matter (i.e., 75% white matter and 25% gray matter). The dielectric measurements were performed according to the method shown in [33]. In particular, as a sensor, an open-ended coaxial probe (RG402) was used and a vector network analyzer TR5048 was adopted. For each measurement, a 5 MHz frequency step in the frequency range from 1 to 2.4 GHz was considered. Moreover, in order to take account of the measurement uncertainties, for each tissue, the measurement was repeated ten times. Accordingly, the measurements depicted in Figure 8 refer to the average value taken from ten measurements for each tissue. As can be seen, the measurement values at 1 GHz are in agreement with the values in Table 3.

The stratified phantom, composed of the skin, bone, CSF, and brain, was carried out by employing three ABS structures printed with a 3D printer [25] (see Figure 9a). The three structures were waterproofed with a resin layer to avoid contamination between the tissues themselves. Then, these were filled with the proper mixture and placed one into the other. The skin was replaced with pig skin that covered the entire phantom. In Figure 9b, the phantom inserted in the helmet is shown. The same set-up adopted for the mixture tissue measurement was adopted to characterize the pig skin, whose value at 1 GHz is indicated in Table 3.

## 5. Algorithm Description

The measurement configuration is multiview/multistatic and multifrequency. The idea is to detect evolving phenomena affecting the brain, such as a hemorrhagic stroke, after the acute phase. Accordingly, the differential strategy proposed in [16] suits us. Overall, the entire measurement was carried out at two time instants, a first measure at time $t_{1}$ and the second one after a certain time, say $t_{2}$. The pneumatic system guarantees that during measurements the device is worn correctly and stably by the patient. This improves the measurement accuracy.

Say $ro_{1},ro_{2},\cdots ,ro_{No}$ are the antenna’s positions and No is the number of antennas. In addition, denote as $r_{1},r_{2},\cdots ,r_{Ns}$ the coordinates of the pixels that divide the spatial region under test *D*. The actual data are represented by the subtraction of signals at the two time instants. Accordingly, the differential scattering matrix, for each frequency, is
(1)$$\Delta S\left(\omega_{i}\right)=H_{r}\left(\omega_{i}\right)A\left(\omega_{i}\right)\delta C\left(\omega_{i}\right)A^{T}\left(\omega_{i}\right)H_{t}\left(\omega_{i}\right)$$
where $H_{r}\left(\cdot \right)$ and $H_{t}\left(\cdot \right)$ take into account the response of the receiving and transmitting antenna, respectively. $A(\omega_{i})$ is the matrix propagator whose *n*-th column has the form $A^{n}\left(\omega_{i}\right)={\left[G\left(\omega_{i},ro_{1},r_{n}\right),G\left(\omega_{i},ro_{2},r_{n}\right),\cdots ,G\left(\omega_{i},ro_{No},r_{n}\right)\right]}^{T}$, and $\delta C(\omega_{i})$ is the N×N differential scattering coefficient correlation matrix. Note that the variation $\delta C\left(\omega_{i}\right)=C_{t_{2}}\left(\omega_{i}\right)-C_{t_{1}}\left(\omega_{i}\right)$ is assumed to be small for using the linear scattering model and is different from zero in the region where the the stroke appears and changes. The operator ${\left(\cdot \right)}^{T}$ indicates the transpose operator.

As described in [34], the MUSIC algorithm is based on the building of a pseudospectrum,
(2)$$\phi \left(\omega_{i},r_{n}\right)=\frac{1}{\left|\right|P_{N}\left[SV^{n}\left(\omega_{i}\right)\right]{\left|\right|}^{2}}$$
that peaks where the stroke appears or changes. This occurs when the so-called noise subspace $P_{N}\left[\cdot \right]$ and the steering vector SV(·) are orthogonal.

In particular, the noise subspace is computed by the eigenspectrum of $\Delta S(\omega_{i})$ and the steering vector is the normalized column of the propagator $A(\omega_{i})$, that is, $SV^{n}\left(\omega_{i},r_{n}\right)=\frac{A^{n}\left(r_{n},\omega_{i}\right)}{\left|\right|A^{n}\left(r_{n},\omega_{i}\right)\left|\right|}$, being evaluated in correspondence to the trial position $r_{n}$ within the spatial domain *D*.

Equation (Equation 2) refers to single-frequency data. Multiple frequencies can be incoherently combined as in [13], which yields
(3)$$\Phi \left(r_{n}\right)=\prod_{i=1}^{N_{f}}\phi \left(\omega_{i}r_{n}\right)$$

All the details of the algorithm can be found in [14]. The final 3D pseudospectrum formation is built by combining 2D pseudospectrums at different heights. Accordingly, the 3D pseudospectrum becomes
(4)$$I\left(r_{n}\right)=\left\{\begin{array}{c} \Phi (r_{n},z_{1})\\ \cdots \\ \Phi (r_{n},z_{H}) \end{array}\right\}$$
where $z_{h}\in \left(z_{1},\ldots ,z_{H}\right)$ are the different heights at which the slices are taken. By exploring the $I(r_{n})$, the detection of the stroke occurrence is achieved.

## 6. Experimental Results

In this section, the diagnostic capability of the TES prototype system (Figure 3) is validated against some experimental scenarios.

First of all, the actual performances of the prototype antenna shown in Figure 4 are checked. In practical cases, this means to ensure a scattering parameter $S_{11}$ level not exceeding −10 dB. This is confirmed in Figure 10, where three cases are considered when a two-layered phantom composed of pig skin and brain material is considered. The position $p_{0}$ refers to the no-load case, where the antenna is in the free-space scenario. Clearly, in this situation, the antenna performances are very poor. When the antenna is posed in $p_{1}$, where a small air-gap between the antenna and the skin layer is considered, the behavior changes positively. Obviously, the design requirement is only partially satisfied, confirming the numerical results in Figure 7. Finally, position $p_{2}$ refers to placing the antenna perfectly on the skin layer. In this case, the design requirement is totally satisfied.

Before testing the stroke detection capability of the system, its stability and accuracy in measuring the attenuation level due to the transmission channel were evaluated. A Monte Carlo analysis was performed for different channel attenuation levels mimicking the different working conditions of the system (i.e., for each single measurement of Monte Carlo analysis, the entire acquisition procedure summarized in Figure 11a and Figure 12a is repeated). In reference to the diagram shown in Figure 11a, an attenuation ranging from α=−15 dB up to α=−65 dB with a step of 10 dB is considered. Figure 11b shows the system performance in terms of both the amplitude and phase content of $S_{12}$. In more detail, each histogram in Figure 11b shows the difference between the actual attenuation value and the one measured that is affected by system noise. As can be seen, the system behavior is stable across the frequency band; a slight performance degradation occurs only for the maximum attenuation level (α=−65 dB). This becomes even worse for the upper-frequency band. A similar analysis is performed by replacing the attenuation block with the complete phantom, as shown in Figure 12a. Now, only two transmission channels are considered. The first is the one between the far antennas (WAs); in particular, one is placed in front of the helmet and the other to its back. The second considers the nearest antennas (NAs) positioned on the same side of the helmet. As can be seen, the results for the WA configuration are in agreement with the ones achieved with α=−55 dB (see Figure 11b). Similarly, the performance for the NA scenario is very close to the case of α=−35 dB (see Figure 11b).

Initially, in order to check the detection capability of the TES prototype, a simple scenario is considered. Thus, the phantom is composed of a single structure covered externally by pig skin. The structure inside is empty and a 30 mm radius plastic balloon filled with a blood mixture is inserted, as shown in Figure 13a. The balloon is located on the slice with the highest number of antennas. The data are collected in the frequency band [1–2] GHz, taking only 50 equi-spased frequencies. In order to establish the performance results obtained by Incoherent MUSIC (MU) against some other approach proposed in the literature, the Incoherent Migration scheme (Mi) is considered [35]. Figure 13 shows the reconstructions achieved via the MU and Mi strategies in different scenarios. In the first column, one of the measures of the differential data is the background measurement, which refers to the no-target case. Panels (d) and (g) show the normalized amplitude reconstructions at the 2D slice corresponding to the maximum of the 3D reconstruction.

In the middle column, the structure is filled with the brain mixture, and the balloon is fixed to a syringe to simulate the variation in volume between two consecutive measurements as shown in Figure 13b. In particular, the target volume passes from 10 mL at first measure $t_{1}$ to 26 mL at second measure $t_{2}$. Here, a hemorrhagic scenario is represented. On the contrary, in the last column, an empty balloon is considered as a target. Of course, this does not intend to mimic a clot situation but aims to show that in principle the TES device can detect anomalies with dielectric properties lower than those in healthy brain tissues, where dielectric contrast is opposite in sign to the hemorrhagic situation. From Figure 13, it can be observed that, under the same scenario, the MU method outperforms the Incoherent Migration one. This is confirmed by looking at Table 4, where the same metrics defined in [36] are used for performance evaluations. In particular, both the incoherent methods have similar focusing capabilities, but MUSIC reaches a larger dynamic range between clutter and target levels than Migration.

In Figure 14, the TES device is validated for more complex scenarios. The complete phantom described in the previous section is considered. Only the hemorrhagic phenomenon is examined and the stroke growth is the same as that in Figure 13b. Figure 14 depicts the stroke reconstructions located at four different positions. In all cases, the targets are detected and localized very well. Moreover, a few artifacts appear in the images which are greatly mitigated thanks to the adopted reconstruction algorithm [37].

Finally, in Table 5 the TES system is compared to other portable microwave prototypes shown in the literature. As can be seen, all systems work in a quasi-similar frequency band, but only our proposed prototype generates the RF signal by adopting a dedicated circuit, avoiding using the vector network analyzer (VNA).

## 7. Conclusions

A portable helmet prototype for brain stroke detection was presented. The stability and accuracy of the system were evaluated with Monte Carlo analysis, showing good performances.

The results of experimental scenarios confirm that the device is able to monitor brain strokes in the post-acute stage. In particular, Incoherent MUSIC was compared with Incoherent Migration, showing better performance in terms of SCR and SMR. Moreover, compared to other similar systems, the proposed device is wearable and allows for an early diagnosis.

Current studies are moving towards the refinement of the pneumatic system to improve the measurement stability needed when the differential strategy is adopted. In addition, the device can be used also for traumatic brain injury (TBI).

## Figures and Tables

**Figure 1 sensors-24-02887-f001:**
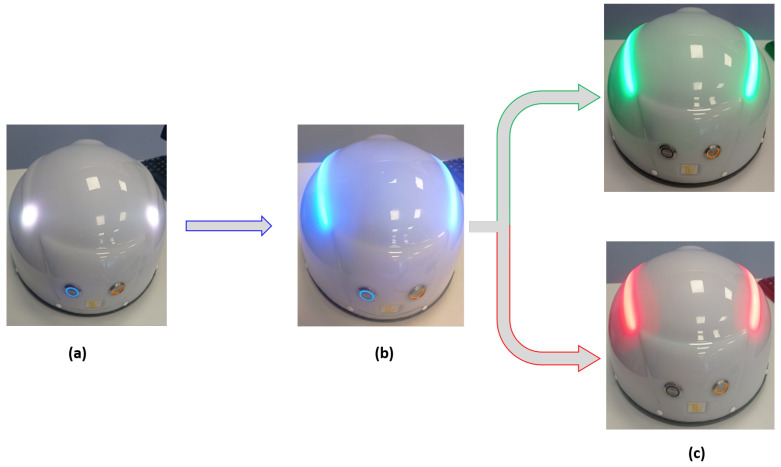
TES prototype in different operative conditions: (**a**) scanning step, (**b**) processing step, and (**c**) processing result.

**Figure 2 sensors-24-02887-f002:**
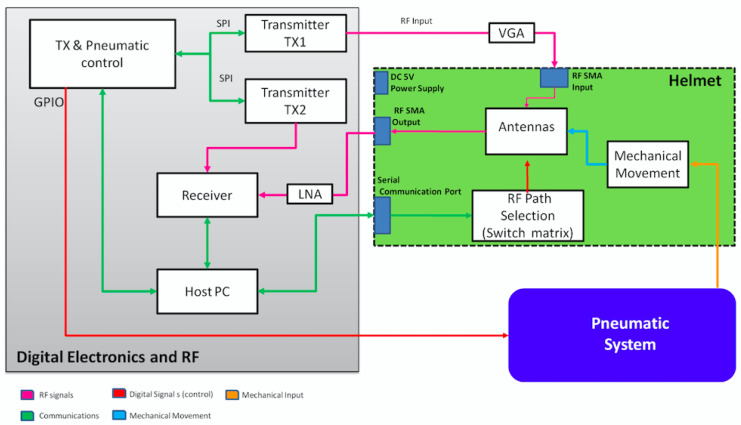
Three-dimensional stroke microwave scanner: block diagram.

**Figure 3 sensors-24-02887-f003:**
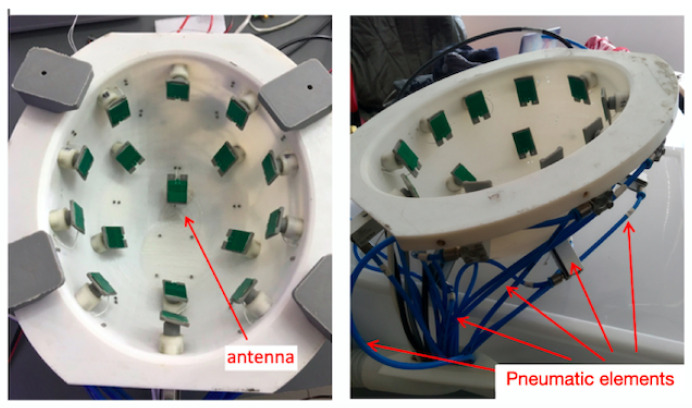
Helmet prototype.

**Figure 4 sensors-24-02887-f004:**
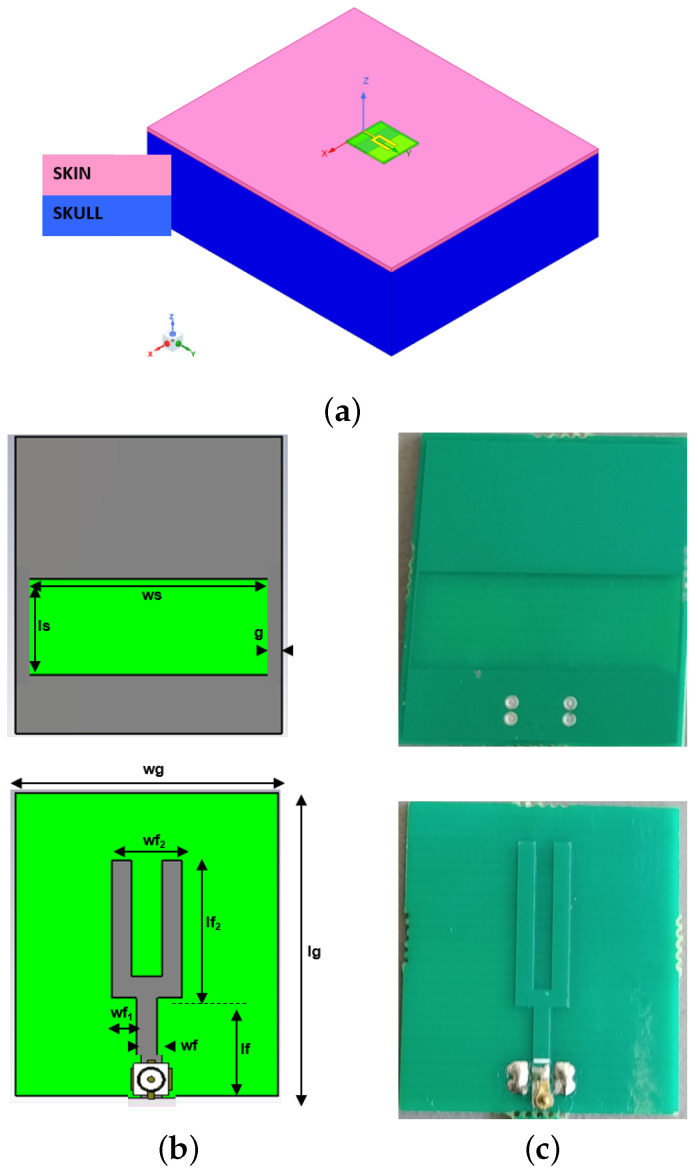
Pictorial view of the simplified two-layered phantom antenna used during the design of the antenna. (**a**) The two-layered planar simplified head phantom; (**b**,**c**) top and bottom views of the antenna design and its prototype.

**Figure 5 sensors-24-02887-f005:**
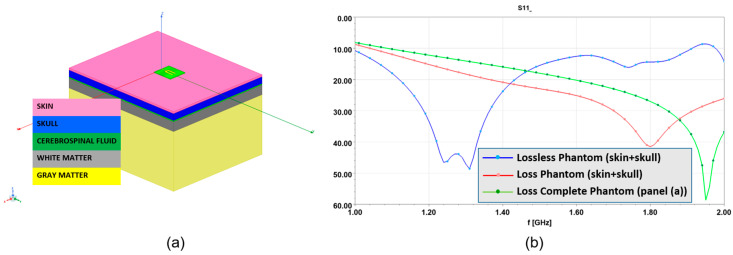
Numerical checking of the designed antenna. (**a**) Five-layered phantom (complete phantom); (**b**) $S_{11}$ behaviors pertaining to the simplified two-layered phantom depicted in Figure 4a without losses (blue line with dots) and with losses (red line with dots), and to the complete phantom of panel (**a**) in this figure.

**Figure 6 sensors-24-02887-f006:**
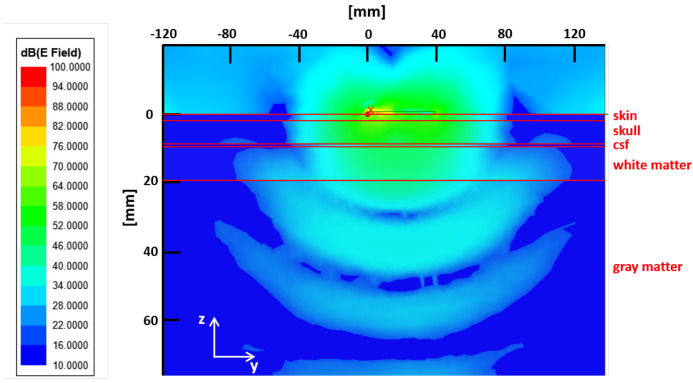
Amplitude field |E| distribution at 1 GHz. The numerical experiment refers to same scenario displayed in Figure 5a.

**Figure 7 sensors-24-02887-f007:**
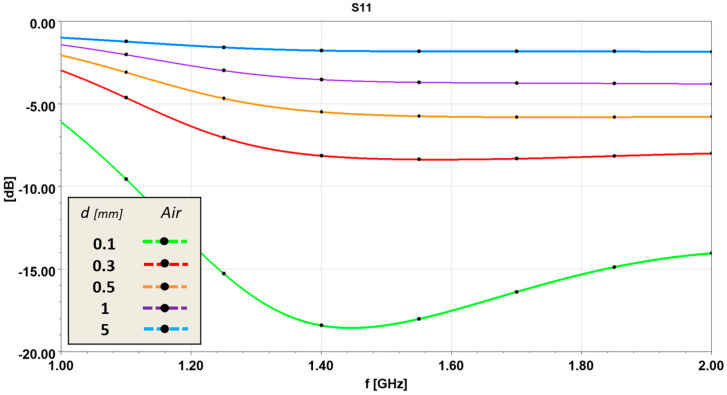
Numerical analyses of Antenna $S_{11}$ for different air-gap circumstances between antenna and skin, considering phantom as in Figure 5a.

**Figure 8 sensors-24-02887-f008:**
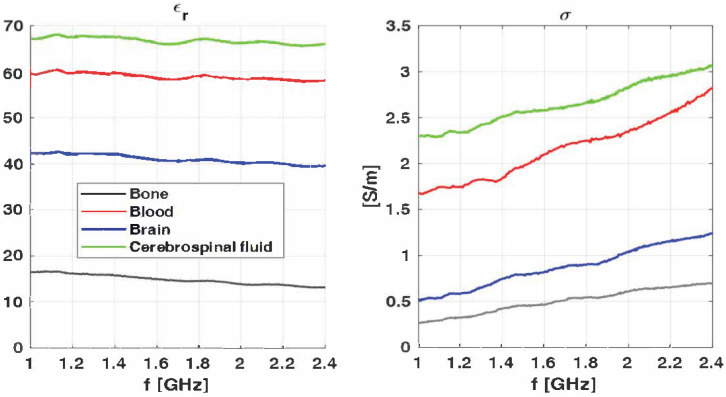
Measurement behaviors of different layers at varying frequencies.

**Figure 9 sensors-24-02887-f009:**
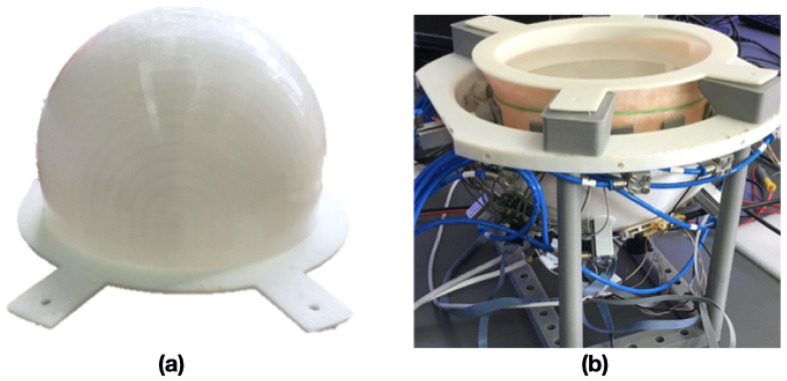
Phantom structure: (**a**) single ABS layer, (**b**) head phantom inserted into the helmet.

**Figure 10 sensors-24-02887-f010:**
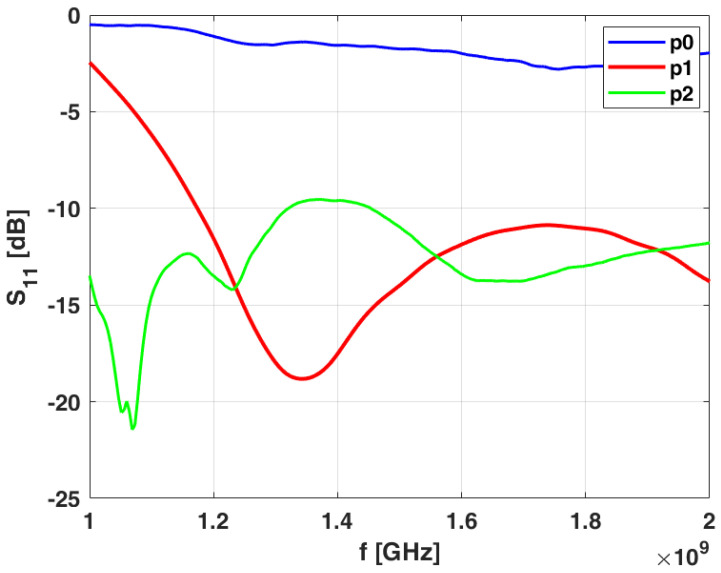
Scattering parameter $S_{11}$ of prototype antenna in three configurations: antenna in air (blue line), antenna touching phantom with air-gap circumstance (red line), and antenna perfectly in contact with phantom (green line). Phantom is composed of only two layers: pig skin and brain material.

**Figure 11 sensors-24-02887-f011:**
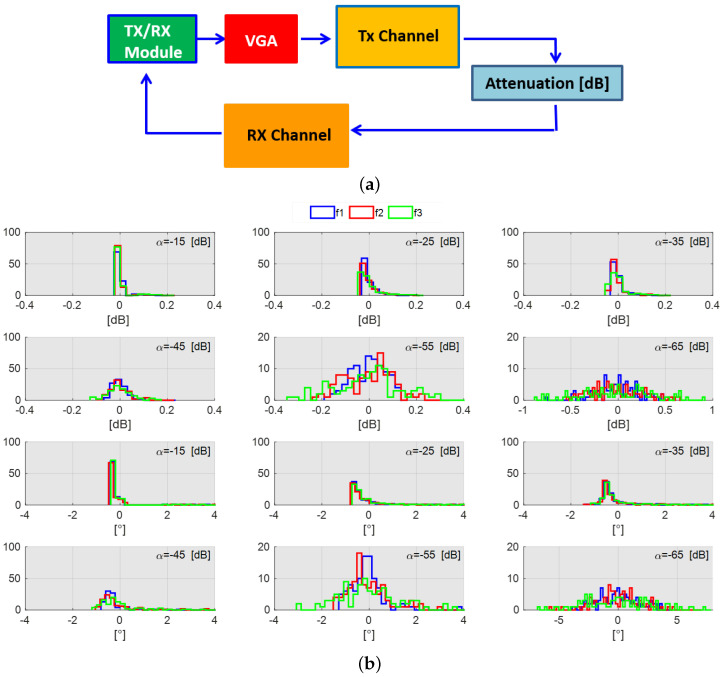
Stability and accuracy analysis for lower ($f_{1}=1$ GHz), middle ($f_{2}=1.5$ GHz), and higher ($f_{3}=2$ GHz) frequencies: (**a**) block diagram of TES system and (**b**) performance analysis.

**Figure 12 sensors-24-02887-f012:**
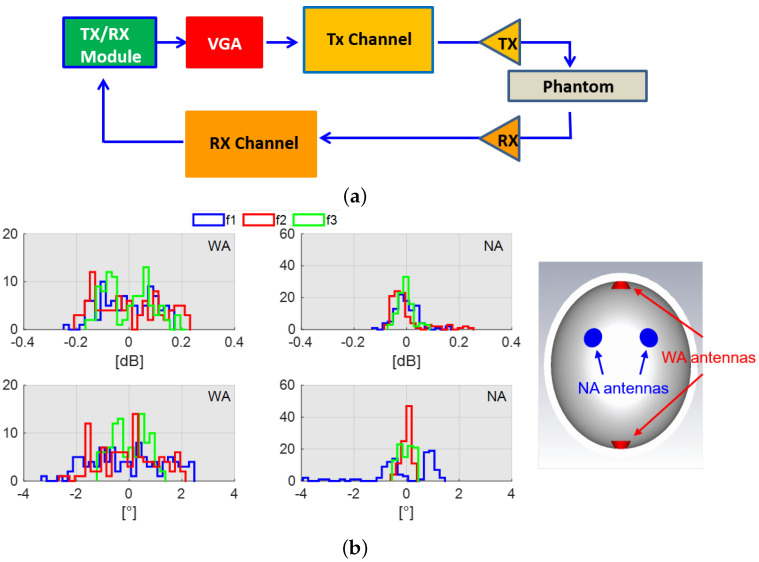
Stability and accuracy analysis with phantom for lower ($f_{1}=1$ GHz), middle ($f_{2}=1.5$ GHz), and higher ($f_{3}=2$ GHz) frequencies: (**a**) block diagram of TES system and (**b**) performance analysis.

**Figure 13 sensors-24-02887-f013:**
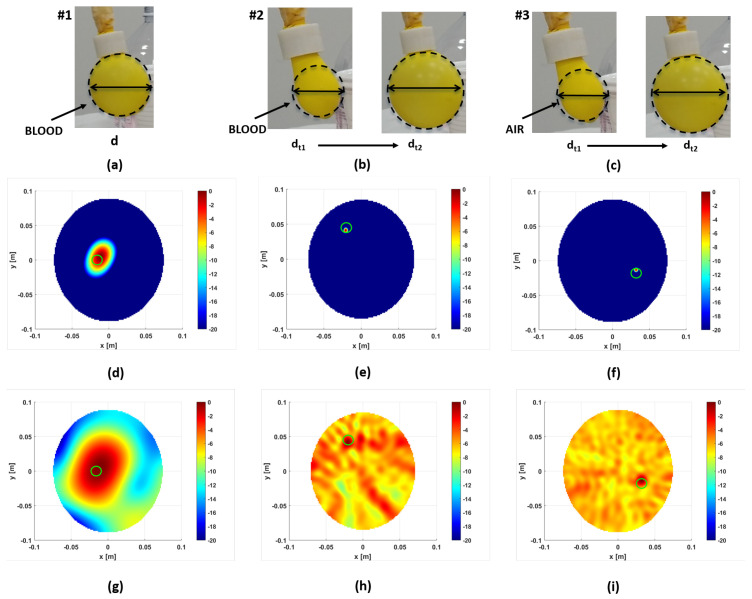
Measurement results: (**a**–**c**) different targets used. Horizontal cross-sections corresponding to 2D slice height where maximum of 3D reconstruction is achieved: (**d**–**f**) Incoherent MUSIC reconstructions and (**g**–**i**) Incoherent Migration reconstructions. Green solid represents actual target.

**Figure 14 sensors-24-02887-f014:**
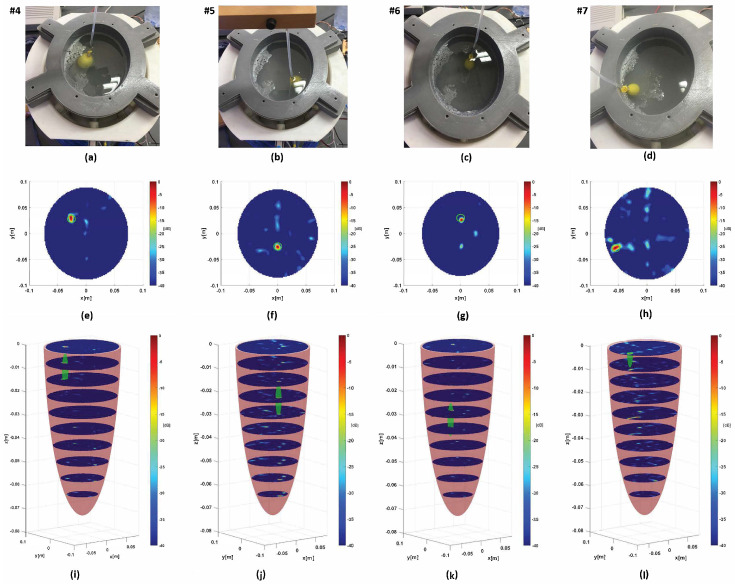
Measurementresults: (**a**–**d**) complete phantom with target in different positions, (**e**–**h**) horizontal cross-sections corresponding to 2D slice height where maximum of 3D reconstruction is achieved, (**i**–**l**) set of 11 2D slice levels at different heights. Green solid represents target.

**Table 1 sensors-24-02887-t001:** Dimensions of designed antenna.

Label	Value [mm]	Label	Value [mm]
wf	0.81	$lf_{2}$	10.00
wg	19.00	lg	20.70
$wf_{1}$	2.21	lf	6.17
$wf_{2}$	5.22	*g*	0.39
ws	18.22	ls	6.93

**Table 2 sensors-24-02887-t002:** Tissue properties of numerical complete phantom shown in Figure 5a at 1 GHz.

Tissue	Skin	Bone	CSF	White Matter	Gray Matter
$\epsilon_{r}$	41	18	68.2	35.58	52.30
σ [S/m]	0.85	0.40	2.30	0.62	0.98
Thickness [mm]	2	7	1	10	50

**Table 3 sensors-24-02887-t003:** Tissue recipes and relative electromagnetic characteristics at 1 GHz.

Tissue	Pig Skin	Bone	CSF	Brain	Blood
$\epsilon_{r}$	41±2	12±1	68±8	42±2	61±3
σ [S/m]	0.53±0.05	0.20±0.07	2.5±0.3	0.71±0.03	1.62±0.08
Triton X-100 [vol %]	/	75	6	38	14
NaCl [g/L]	/	0.8	13.7	5.2	9.4

**Table 4 sensors-24-02887-t004:** Reconstruction metrics: signal to clutter ratio (SCR), signal to mean ratio (SMR), and spatial displacement (SD).

Experiment	SCR [dB]		SMR [dB]		SD [mm]	
	MU	Mi	MU	Mi	MU	Mi
#1	5.10	5.67	43.40	11.18	6.40	7.10
#2	25.22	1.54	72.30	5.95	3.60	3.60
#3	23.32	3.85	79.35	6.3	4.60	4.66
#4	14.27	1.60	65.21	5.73	2.03	2.05
#5	24.65	1.87	54.50	5.02	5.02	5.7
#6	18.36	1.14	69.01	5.88	13.17	13.23
#7	5.7	1.32	50.90	4.67	8.9	8.97

**Table 5 sensors-24-02887-t005:** TES system compared to other portable device shown in the literature.

Portable System	Frequency Band	Exam Duration [min]	Coupling Medium	Processing Load	RF Signal Generation System
[4]	[0.3–3] GHz	ND	✓	Medium	VNA
[9]	[1.2–4] GHz	ND	*x*	Low	VNA
[10]	[0.1–1.95] GHz	3	✓	Medium	VNA
[11]	[0.5–2.5] GHz	ND	✓	Medium	VNA
TES	[1–2] GHz	4	*x*	Low	RF circuit dedicated

## Data Availability

Data is contained within the article.

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
