# Peer review of "Subcranial Encephalic Temnograph-Shaped Helmet for Brain Stroke Monitoring"

_sensors, 2024, doi:10.3390/s24092887_

Round 1
Reviewer 1 Report
Comments and Suggestions for Authors
I don't understand figure number 5. The graph doesn't describe the y-axis (I think it should be in negative numbers). Although the five-layer phantom is shown in Fig. 5 (a), but from legend it seems that S11 is only shown for skin+skull. I did not find out how the lossless phantom was created.
Why the experimental verification of S11 in Figure 10 does not agree at all with the numerical simulations in Figure 5(b) or Figure 6.
For Figure 10 Specify how the perfect contact with the phantom was created when the antennas are planar and the phantom is spherical. I would appreciate the S11 for a real system and phantom scenario.
Please include the type of probe used for dielectric parameters measurement, including the VNA manufacturer. I found a VNA TR1300 that measures up to 1.3 GHz, but on Figure 8 results are presented for up to 2.4 GHz.
From Table 3, it appears that the measured dielectric parameters are accurate. It would be appropriate to state the measurement uncertainties.
Since the helmet is portable, a comparison with portable devices for stroke type classification could be made.
Reviewer 2 Report
Comments and Suggestions for Authors
I think the system is well designed and well described.
Just some minor edits are required before publication:
1 - Authors state that they are using measurements taken at different times. At this point, it is better to put an evolutional analysis, where the bleeding starts from 0 and reaches the max point in 3-4 steps and the reverse of the same process. Can proposed algorithm identify the evolution (is it a shrinkage or expansion?), how they will use these sequential differences to understand the direction of evolution.
2 - Please check all figs and equations:
In Fig 7, there is no dimension on axes (cm or mm)?
In eq (2), "SVn(ωi" parenthesis is missing.
Round 2
Reviewer 1 Report
Comments and Suggestions for Authors
1)Measurement uncertainties should be inclued. At least type A uncertainties from 10 measurements, but I believe that type B uncertainties can also be determined with a commercially available measurement set and present an expanded type C uncertainty. Mixture may not always be homogeneous. This is how the results look accurate. Uncertainties would contribute to comparisons with future works.
2)The comparison was made with literature that is more than 3 years old, which is often followed by newer works by the authors.
3) Figure 14 and Tablets 4 and 5 are pointlessly incorporated into the references.
5) I recommend listing devices in the text in the following format: name, manufacturer, country
Author Response
1)Measurement uncertainties should be inclued. At least type A uncertainties from 10 measurements, but I believe that type B uncertainties can also be determined with a commercially available measurement set and present an expanded type C uncertainty. Mixture may not always be homogeneous. This is how the results look accurate. Uncertainties would contribute to comparisons with future works.
Response 1: Thanks for the suggestion. In Table 3 uncertainties were added in the revised version of the paper.
2)The comparison was made with literature that is more than 3 years old, which is often followed by newer works by the authors
Response 2: We thank the reviewer for his observation. In the comparison, we considered only portable devices, excluding laboratory devices.
3) Figure 14 and Tablets 4 and 5 are pointlessly incorporated into the references.
Response 3: Thanks for the remark. Figure 14 and Tables 4 and 5 were moved up.
4) I recommend listing devices in the text in the following format: name, manufacturer, country
Response 4: We thank the reviewer but, unfortunately, the information requested by the authors is not present in all the references cited in Table 5.